# Long-term durability of metastable β-Fe₂O₃ photoanodes in highly corrosive seawater

Changhao Liu[1,2], Ningsi Zhang[1,2], Yang Li[1], Rongli Fan[1], Wenjing Wang[1], Jianyong Feng[1] ✉, Chen Liu[3], Jiaou Wang [3], Weichang Hao [4], Zhaosheng Li [1,2] ✉ & Zhigang Zou [1,2]

Durability is one prerequisite for material application. Photoelectrochemical decomposition of seawater is a promising approach to produce clean hydrogen by using solar energy, but it always faces the problem of serious Cl⁻ corrosion. We find that the main deactivation mechanism of the photoanode is oxide surface reconstruction accompanied by the coordination of Cl⁻ during seawater splitting, and the stability of the photoanode can be effectively improved by enhancing the metal-oxygen interaction. Taking the metastable β-Fe₂O₃ photoanode as an example, Sn added to the lattice can enhance the M−O bonding energy and hinder the transfer of protons to lattice oxygen, thereby inhibiting excessive surface hydration and Cl⁻ coordination. Therefore, the bare Sn/β-Fe₂O₃ photoanode delivers a record durability for photoelectrochemical seawater splitting over 3000 h.

The use of PEC water splitting to produce hydrogen can realize the conversion of solar energy to hydrogen energy in one step, which is a very promising solution for building a low-carbon society[1–5]. The long-term stability of photoelectrodes is an essential prerequisite for the practical application of PEC water splitting for hydrogen production[6]. However, except for iron oxide, almost all bare photoelectrodes show unsatisfactory stability in water splitting for hydrogen production, let alone in highly corrosive seawater[7–10]. Some strategies, such as protective layers, electrocatalysts, and tuning electrolyte composition, have been used to improve the durability of photoelectrodes in aqueous electrolytes without Cl⁻ ions[11–13]. Little attention has been given to improving the stability of photoelectrodes in aqueous electrolytes with Cl⁻ ions[14,15], since Cl⁻ ions easily corrode photoelectrode materials and may participate in the competitive oxidation reaction to produce Cl₂ or ClO⁻[16–19].

Herein, we studied the effect of Cl⁻ ions on the stability of a photoelectrode such as β-Fe₂O₃. Recently, β-Fe₂O₃ entered our research field as a metastable phase of iron oxide. Because of its narrower band gap (1.9 eV) compared with α-Fe₂O₃ (2.1 eV), the theoretical

optical absorption band edge can be extended to approximately 650 nm. Thus, it has a higher theoretical solar-to-hydrogen efficiency than α-Fe₂O₃. At the same time, β-Fe₂O₃ also shows good stability in photoelectrochemical alkaline water splitting[20,21]. We have revealed that the Cl⁻ ions in seawater will damage the surface hydrated layer of β-Fe₂O₃ photoanodes, thus remarkably reducing their stability. Dispersed Sn single atoms in the lattice were found to endow the β-Fe₂O₃ photoanodes with good inhibition of hydration and resistance to Cl⁻ attack in seawater. As a result, Sn/β-Fe₂O₃ without any protective overlayer shows excellent durability in seawater splitting over 3000 h and is by far the most stable photoanode. This study may ignite the dawn of application for PEC seawater splitting for hydrogen production and deepen the understanding of the seawater corrosion of oxides.

## Results

Metastable β-Fe₂O₃ photoanodes doped with Sn were prepared by the spray pyrolysis method, and their phases were accurately determined (Supplementary Fig. 1). The Sn/β-Fe₂O₃ film is composed of blocks

[1]Collaborative Innovation Center of Advanced Microstructures, National Laboratory of Solid State Microstructures, College of Engineering and Applied Sciences, Nanjing University, 22 Hankou Road, Nanjing 210093, China. [2]Jiangsu Key Laboratory for Nano Technology, Nanjing University, 22 Hankou Road, Nanjing 210093, China. [3]Beijing Synchrotron Radiation Facility, Institute of High Energy Physics, Chinese Academy of Sciences, Beijing 100049, China. [4]School of Physics and Centre of Quantum and Matter Sciences, International Research Institute for Multidisciplinary Science, Beihang University, Beijing 100191, China. ✉e-mail: fengjianyong@nju.edu.cn; zsli@nju.edu.cn

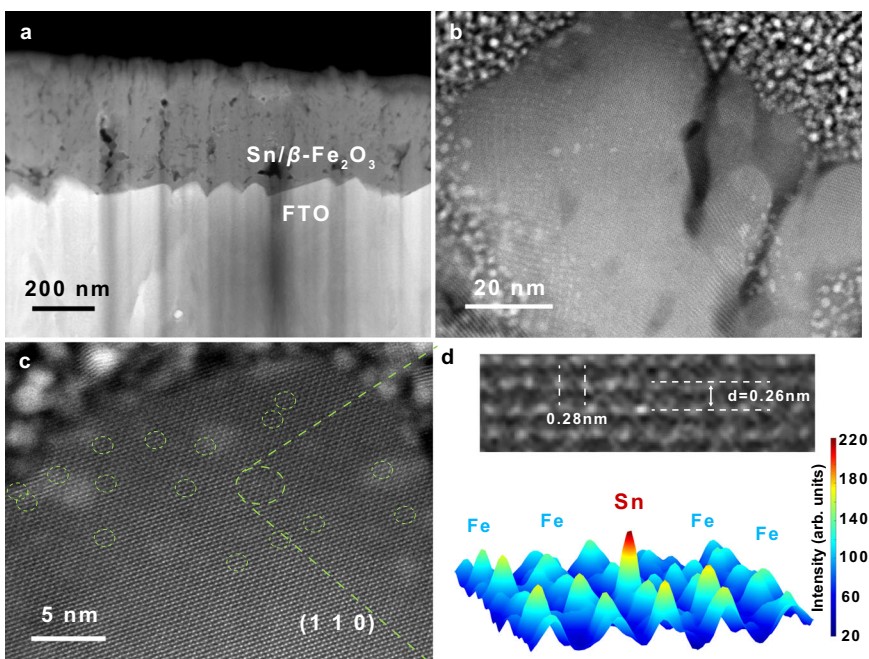

**Fig. 1 | Cross section and crystal structure of the Sn/$\beta$-Fe$_2$O$_3$ photoanode.**
**a, b** HAADF images of the Sn/$\beta$-Fe$_2$O$_3$ film cross-section at different magnifications. **c** Atomic image of the (1 1 0) plane of $\beta$-Fe$_2$O$_3$. **d** Local enlargement near doped Sn atoms and three-dimensional modelling of surface contrast.

arranged vertically with a thickness of approximately 400 nm (Fig. 1a). A large area of lattice stripes indicates good crystallinity of $\beta$-Fe$_2$O$_3$ (Fig. 1b). Many bright spots with high contrast in the (1 1 0) crystal plane in Fig. 1c correspond to the Sn single atom in the $\beta$-Fe$_2$O$_3$ lattice. One of the regions was selected for three-dimensional modelling, which shows the contrast difference between Sn atoms and surrounding Fe atoms (Fig. 1d). It was confirmed that the lattice position of Fe was substituted by Sn. A clear atomic image of the (1 1 1) crystal plane taken from another region of $\beta$-Fe$_2$O$_3$ and fast Fourier transform (FFT) patterns of the (1 1 0) and (1 1 1) planes were obtained (Supplementary Fig. 1). These lattice atomic images and FFT patterns are completely consistent with the atomic arrangement of the corresponding crystal plane in the theoretical model.

The as-prepared $\beta$-Fe$_2$O$_3$ photoanodes were tested for PEC simulated seawater splitting. The photocurrent density of the 2% Sn/$\beta$-Fe$_2$O$_3$ photoanodes reaches 2.21 mA cm$^{-2}$ at 1.6 V$_{RHE}$, which is 8.5 times that of $\beta$-Fe$_2$O$_3$ photoanodes (Supplementary Fig. 2). The Sn dopants do not affect the light absorption of the $\beta$-Fe$_2$O$_3$ photoanodes, while the PEC performance improvement is partly due to the increased electron concentration caused by the doping of high-valence cation Sn$^{4+}$. The electron concentration of Sn/$\beta$-Fe$_2$O$_3$ is approximately 8.4 times that of $\beta$-Fe$_2$O$_3$, and correspondingly, its bulk conductivity is also improved, according to Mott-Schottky plots and AC electrochemical impedance spectra in the low-frequency region. Sn can simultaneously adjust the chemical field at the semiconductor/electrolyte interface, which significantly reduces the AC impedance of the interface transfer kinetics (Supplementary Figs. 3, 4 and 5).

The most important role of Sn dispersed in the lattice is to promote its durability in simulated seawater with Cl$^-$ ions. Specifically, in Fig. 2a, the stability of the $\beta$-Fe$_2$O$_3$ photoanodes is good in 1 M KOH electrolyte within 100 h, while its photocurrent density decreased obviously in 1 M KOH + 0.5 M NaCl electrolyte. This indicates that Cl$^-$ significantly reduced its PEC stability. In contrast, the Sn/$\beta$-Fe$_2$O$_3$ photoanodes could still maintain stable performance even in a saturated NaCl electrolyte within 100 h without decay, which was much better than the $\beta$-Fe$_2$O$_3$ photoanodes. In the stability test, the photocurrent increased slightly in a period of time after the beginning of the

reaction due to the change in the state of Fe and O on the surface[22–24]. Correspondingly, the AC impedance of the Sn/$\beta$-Fe$_2$O$_3$ photoanode in the first 50 h gradually decreases (Fig. 2b). The HADDF image of the photoanode also shows that an amorphous hydrated layer was formed on the surface of $\beta$-Fe$_2$O$_3$ (Fig. 2c), which indicates that the FeOOH hydrated layer spontaneously formed on the surface during the reaction process. The specific process and impact of surface reconstruction will be further discussed below. Furthermore, the Sn/$\beta$-Fe$_2$O$_3$ photoanode shows excellent stability over 3000 h in simulated seawater (Fig. 2d). After 3000 h, the photocurrent maintains 96.8% of the initial value. The Sn/$\beta$-Fe$_2$O$_3$ photoanode has achieved the longest durability in research on PEC seawater splitting over the years, as shown in Fig. 2e, even without any extra electrocatalyst or protective overlayer. Additionally, the Sn/$\beta$-Fe$_2$O$_3$ photoanodes showed excellent stability in alkaline natural seawater (Supplementary Fig. 6).

The evolution process of the $\beta$-Fe$_2$O$_3$ photoanode surface in seawater splitting is explored here. XPS analysis can be used to obtain the state of the $\beta$-Fe$_2$O$_3$ photoanode surface elements in contact with the electrolyte during the reaction. As shown in the XPS spectrum of O in the $\beta$-Fe$_2$O$_3$ photoanode (Supplementary Fig. 7a), the peak of M–O at 529.4 eV decreases after the reaction. It transforms to M–OH at 531.7 eV, with a significant shift from lattice oxygen to hydroxyl oxygen on the surface[25]. This corresponds to the reconstruction of the $\beta$-Fe$_2$O$_3$ surface in alkaline electrolytes. In the XPS plot of Sn/$\beta$-Fe$_2$O$_3$ photoanodes with 100 h of reaction in Fig. 3a, a large number of O atoms can remain in the form of lattice oxygen when Sn is present on the surface. After 100 h of reaction, no Cl signal was detected on the surface of the Sn/$\beta$-Fe$_2$O$_3$ photoanodes (Fig. 3b). In contrast, the Cl signal was detected on the $\beta$-Fe$_2$O$_3$ surface (Supplementary Fig. 7b), which indicates that the process of lattice oxygen reconstruction was accompanied by the adsorption or implantation of Cl$^-$ in the electrolyte. During surface hydration and lattice reformation, $\beta$-Fe$_2$O$_3$ slowly dissolves, which can be determined by inductive coupled plasma emission spectrometer (Supplementary Table 2). The amount of dissolved Fe atoms was also significantly reduced when Sn acted as an anchor at the surface. However, due to the intense hydration, the surface lattice was still continuously attacked after a long reaction time, accompanied

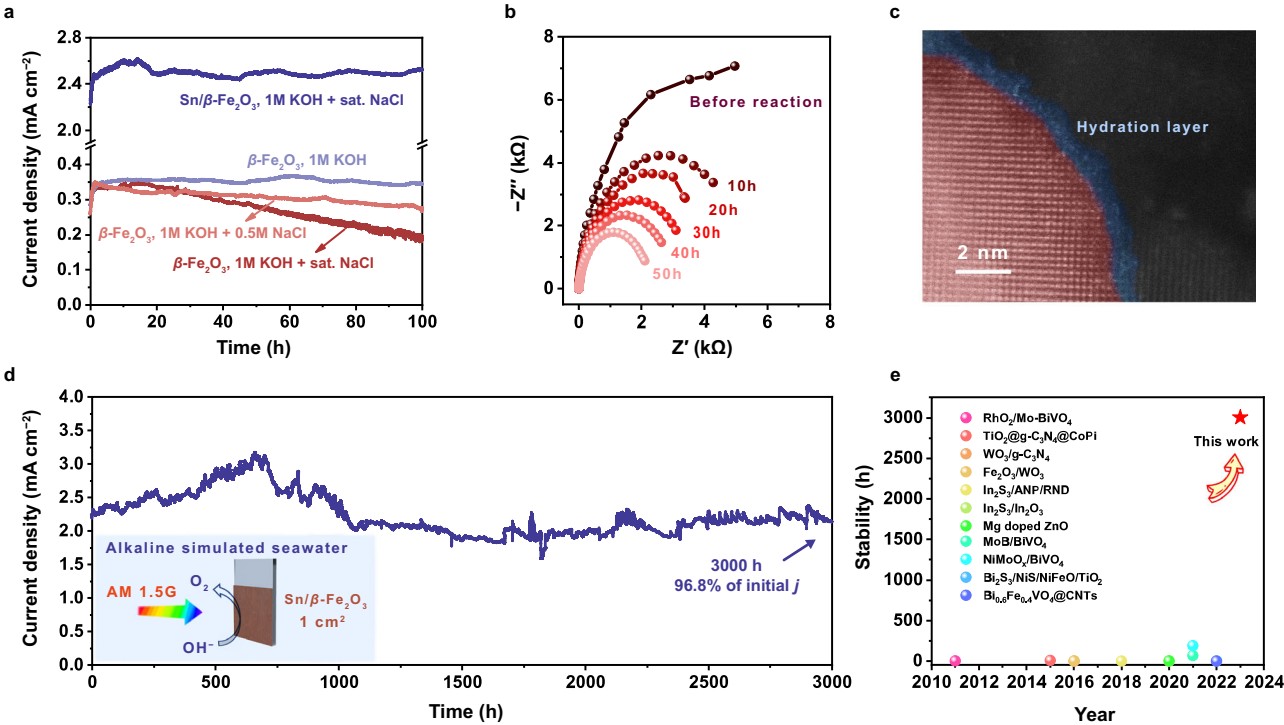

**Fig. 2 | PEC properties of the β-Fe₂O₃ photoanode in simulated seawater.**
**a** Stability test of β-Fe₂O₃ (in 1 M KOH, 1 M KOH/0.5 M NaCl, 1 M KOH/saturated NaCl solution) and Sn/β-Fe₂O₃ (in 1 M KOH/saturated NaCl solution) for 100 h. **b** AC electrochemical impedance spectra of Sn/β-Fe₂O₃ before and after the reaction. **c** HAADF images of the Sn/β-Fe₂O₃ photoanode after 100 h of reaction in 1 M KOH/ 0.5 M NaCl. **d** Stability test of Sn/β-Fe₂O₃ in 1 M KOH/0.5 M NaCl for 3000 h. **e** Summary of the photoanode stability of PEC (simulated) seawater splitting over the years. Detailed information can be found in Supplementary Table 1.

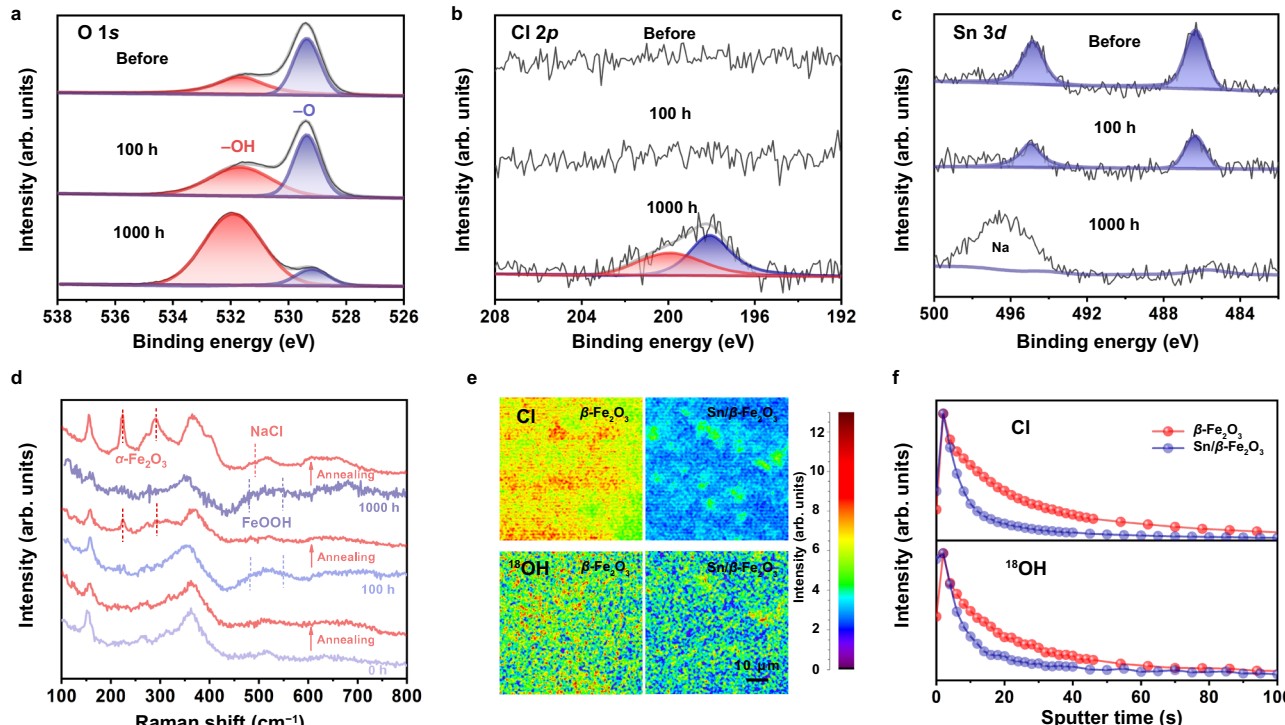

**Fig. 3 | Evolution of the β-Fe₂O₃ surface during the long-term seawater splitting reaction. a**–**c** XPS spectra of O 1s, Cl 2p, and Sn 3d of Sn/β-Fe₂O₃ before, after 100 h, and after 1000 h of seawater splitting reaction. **d** Raman spectra of β-Fe₂O₃ photoanodes with different reaction times and annealing treatments. **e**, **f** TOF-SIMS of the distributions of Cl and ¹⁸OH on the surface and depth profiling of the β-Fe₂O₃ photoanode before and after Sn doping after the 100-h reaction in simulated seawater with 20 wt% H₂¹⁸O.

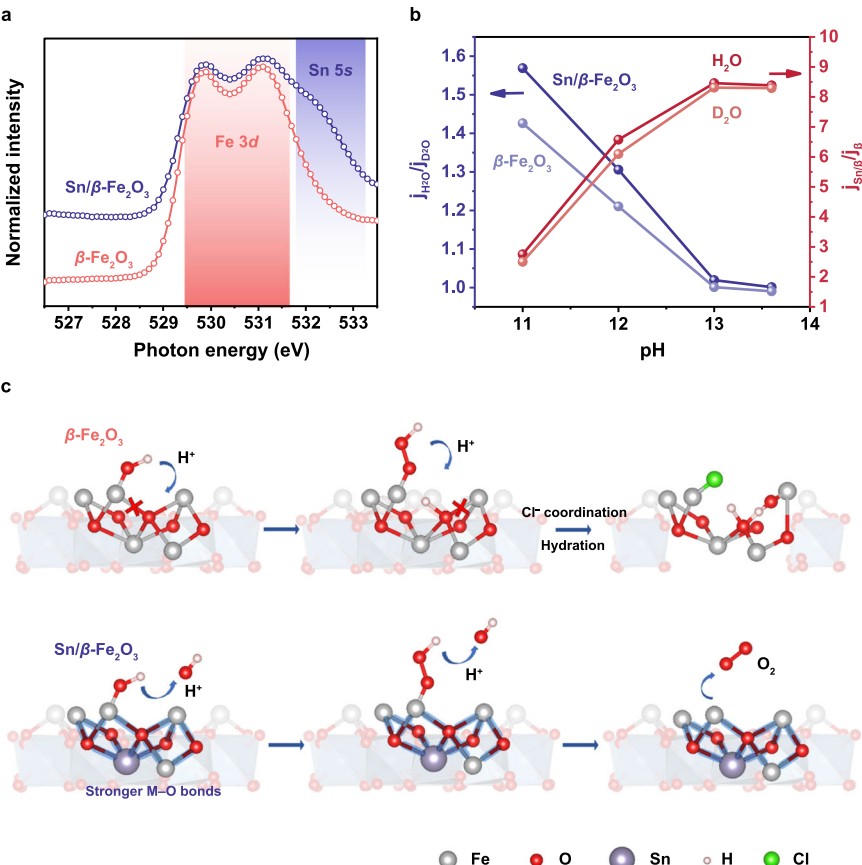

**Fig. 4 | Effect of Sn atoms in $\beta$-Fe$_2$O$_3$ on resistance to seawater corrosion.**
**a** XANES spectra of the O K-edge in $\beta$-Fe$_2$O$_3$. **b** The steady-state photocurrent $j_{H_2O}$/
$j_{D_2O}$ and $j_{Sn}$/$j_{Pure}$ values at different pH values. **c** Schematic diagram of doped Sn
atoms against Cl$^-$ corrosion in seawater splitting. Sn enhances the M−O bonds,
prevents the hydrated surface reconstruction caused by the transfer of H$^+$ to lattice
oxygen, and weakens the coordination of Cl$^-$.

by O remodelling and loss of metal elements. The XPS peak of the Sn
3d signal disappeared after 1000 h of reaction (Fig. 3c), indicating that
Sn is also slowly lost during lattice reconstruction by surface hydra-
tion. Cations are also involved in surface hydration and embedded in
the hydrated layer, such as Na$^+$, and after 1000 h, cations were also
detected on the photoanode surface. Confocal Raman spectroscopy
was used to observe the speciation evolution of trace substances on
the surface of the photoanodes before and after the reaction (Fig. 3d).
After 100 h and 1000 h of seawater splitting, there are two peaks of
M−OOH at approximately 470 and 550 cm$^{-1}$ [26,27], which echo the
change in the O 1s XPS peak. FeOOH, which is hydrated and recon-
stituted during the reaction, is also more prone to dehydration and
sintering at high temperatures. The $\alpha$-Fe$_2$O$_3$ peak can be observed
when the $\beta$-Fe$_2$O$_3$ photoanodes are further calcined at 600 °C. Here, $\alpha$-
Fe$_2$O$_3$ was transformed from FeOOH generated by surface recon-
struction during heat treatment. A NaCl peak appeared on the surface
after 1000 h of reaction, indicating that the crystallization of anions
and cations diffused into the hydration layer.

To further confirm the reconstruction of the $\beta$-Fe$_2$O$_3$ photo-
anodes and the exchange of atoms at the interface during the reaction
in the electrolyte, the surface element distribution measurement was
probed by time-of-flight secondary ion mass spectrometry (TOF-
SIMS). In Fig. 3e, the signal of Cl can be detected on the $\beta$-Fe$_2$O$_3$ surface
after the reaction in simulated seawater for 100 h. The Cl content of the
$\beta$-Fe$_2$O$_3$ surface is much higher than that of the Sn/$\beta$-Fe$_2$O$_3$ surface.
This confirmed that the presence of Sn can significantly improve the
rejection of Cl$^-$ in the electrolyte. Meanwhile, H$_2$$^{18}$O was added to
explore electrolyte participation in the surface reconstruction of the $\beta$-
Fe$_2$O$_3$ photoanode. $^{18}$O in the electrolyte participates in the formation

of a hydration layer, so the signal of $^{18}$OH with surface $m/z$ = 19.005 can
be detected. The signal intensity of $^{18}$OH on the Sn/$\beta$-Fe$_2$O$_3$ photo-
anode surface is much weaker than that on the $\beta$-Fe$_2$O$_3$ surface, indi-
cating that the Sn dopants weaken lattice oxygen reconstruction. In
the depth profiling in Fig. 3f, the content of both Cl and $^{18}$OH in the Sn/
$\beta$-Fe$_2$O$_3$ surface decays faster with depth than without Sn dopants.
This reveals that surface reconstruction and Cl$^-$ erosion occur
simultaneously. The $^{18}$O added to the electrolyte participates in the
reconstruction of the lattice oxygen of $\beta$-Fe$_2$O$_3$ and forms M−$^{18}$OH.
At the same time, Cl$^-$ in the electrolyte would also first be
adsorbed on the surface and gradually infiltrate into the bulk
with surface reconstruction. The $\beta$-Fe$_2$O$_3$ photoanode undergoes
excessive surface reconstruction, resulting in a thicker hydrated layer.
Cl$^-$ shuttles and infiltrates into it, which may destroy the structure of
the $\beta$-Fe$_2$O$_3$ photoanode and affect the interface water oxidation
reaction. Sn inhibits the exchange of $^{18}$O and lattice oxygen and sup-
presses the erosion of Cl$^-$, thus obtaining a more stable photoanode
surface.

The O K-edge of the soft X-ray absorption near-edge structure
(XANES) spectrum shows the change in the lattice oxygen state before
and after adding Sn to the lattice (Fig. 4a and Supplementary Fig. 8).
The spectrogram of $\beta$-Fe$_2$O$_3$ is similar to the O K-edge of standard iron
oxide[28]. After adding Sn, the X-ray absorption peak of oxygen shifts to
the direction of high energy, which reflects that the addition of Sn
effectively improves the bonding energy of O in the lattice. A shoulder
peak at 532.3 eV corresponds to the contribution of the Sn 5s orbit[29,30].
This shows that the Sn atoms dispersed in the lattice change the
average chemical environment of O and play an anchor role in lattice
oxygen.

The enhanced metal–oxygen interaction in the surface chemical reaction is specifically manifested in that the lattice oxygen at the semiconductor electrolyte interface has more difficulty accepting protons, which can be confirmed by the proton-coupled electron transfer process analysed by the H/D kinetic isotope effect[31–33]. The OER on the photoanode surface is a proton-coupled electron transfer process involving four electrons, as shown in Fig. 4b. Specifically, the reaction intermediate species *OH and *OOH transfer one electron to the semiconductor and discard one proton[34,35]. The isotope effect is particularly significant at low pH. The $j_{H2O}/j_{D2O}$ value of the $\beta$-Fe$_2$O$_3$ photoanode is always lower than that of Sn/$\beta$-Fe$_2$O$_3$, indicating that the $\beta$-Fe$_2$O$_3$ surface has a stronger affinity for protons. The lattice oxygen on the surface of $\beta$-Fe$_2$O$_3$ easily acts as a proton acceptor, which to some extent accelerates the proton coupling process in the reaction process. However, lattice oxygen as a proton acceptor will bring about the problem of structural stability being destroyed. As demonstrated in Fig. 4c, protons transferred to nearby locations will combine with lattice oxygen, break the M−O bond, and generate an FeOOH hydrated layer. When the M−O bond breaks, oxygen in solution will exchange with lattice oxygen, and Cl$^-$ will also coordinate with Fe and destroy the surface structure. On the other hand, hydrated FeOOH is a loose amorphous or layered structure, which is also prone to the insertion and adsorption of Cl$^-$, thus affecting the activity of water splitting. The Sn atoms dispersed in the lattice play a role in anchoring the lattice oxygen to prevent proton coupling between the reaction intermediate and the lattice oxygen. which shows that the proton transfer process will have a greater impact on the reaction kinetics. When the pH rises, proton transfer is no longer the rate-determining step. The advantages of donor Sn$^{4+}$ dispersed in the bulk phase in improving electron concentration and conductivity can also be fully demonstrated. Therefore, the photocurrent increases to 8.5 times that of the $\beta$-Fe$_2$O$_3$ photoanode. Although alkaline electrolytes are used in PEC tests, local pH will decrease in the water oxidation reaction, and protons with higher local concentrations will also exist. These protons attack lattice oxygen, causing surface reconstruction. Sn in the lattice enhances the metal−oxygen interaction, thus inhibiting the wrong proton transfer path and avoiding surface hydration and Cl$^-$ corrosion.

## Discussion

The advantage of uniformly dispersed Sn in the bulk phase is that when the surface is hydrated and peeled by corrosion, the exposed Sn/$\beta$-Fe$_2$O$_3$ is still corrosion resistant. During the annealing process of the Sn/$\beta$-Fe$_2$O$_3$ photoanode, Sn atoms tend to diffuse to the surface[36,37], but the Sn in the bulk is still relatively uniform (Supplementary Fig. 9). Such characteristics make the stability rise continuously at first, then decline slowly, and finally maintain a stable range of fluctuations. At the initial stage of the reaction, the $\beta$-Fe$_2$O$_3$ on the surface is converted into FeOOH in situ (Fig. 3). FeOOH itself is an efficient electrocatalyst, so the photocurrent increased. Because of the infiltration corrosion of Cl$^-$ along with the excessive surface reconstruction and the loss of Sn, the photocurrent density had a downwards trend after 700 h. It can be analysed from Supplementary Fig. 10 that due to the electrocatalytic effect of surface FeOOH after the 1000-h reaction, the onset potential moved to the negative direction by approximately 0.1 V$_{RHE}$. However, the photocurrent density at 1.6 V$_{RHE}$ was reduced with the influence of surface reconstruction and corrosion. Owing to the relatively uniform distribution of Sn in the whole film, reconstruction, and corrosion were controlled within a certain range of the $\beta$-Fe$_2$O$_3$ surface instead of continuing to the deep layer. In addition, while the surface reconstruction was accompanied by the loss of Sn, there would also be a sedimentation equilibrium on the surface, and deposition would occur, which forms a dynamic balance of corrosion, metal loss, deposition, and protection. A dynamic stable state of surface evolution was established after a period of time, so the photocurrent density tended to be stable after 1000 h. In contrast, we covered a layer of

efficient OER electrocatalyst CoFe-LDH on the surface of Sn/$\beta$-Fe$_2$O$_3$ as a protective passivation layer. However, a significant downwards trend of the photocurrent was observed in the first 50 h of the reaction, and after a long time, the current gradually decreased to the level without loading the electrocatalyst (Supplementary Fig. 12). This shows that the surface modification of the electrocatalyst cannot resist the corrosion of Cl$^-$ in seawater. Metal hydroxide itself will also be reconstructed in the OER reaction, which will also be accompanied by the problems of Cl$^-$ coordination and structural collapse. Finally, the structure is destroyed and gradually dissolved and peeled off. This extra loaded electrocatalyst protective layer often protects the photoanode by its own corrosion and consumption, which cannot fundamentally solve the long-term stability problem.

In summary, we revealed that excessive hydration reconstruction of the surface will corrode the surface of the oxide photoanode with the corrosion of Cl$^-$ ions in the solution. The anchoring of the surface lattice by Sn hinders the transfer of protons to lattice oxygen, and the probability of oxygen hydrogen bonding will decrease due to the strong M−O bond, thereby suppressing the surface reconstruction and coordination of Cl$^-$. The Sn/$\beta$-Fe$_2$O$_3$ photoanode constitutes by far the most durable photoanode for seawater splitting. This strategy can also improve the durability of other photoanodes, such as $\alpha$-Fe$_2$O$_3$ (Supplementary Fig. 13). This study will pave a new path to solving the problem of the long-term durability of photoelectrodes in energy conversion.

## Methods
### Preparation of photoanode

Typically, in an experiment, 0.01 mol of iron acetylacetonate (AcAcFe) was dissolved in 500 mL ethanol by stirring with a magnetic force for over 48 h. Fluorine-doped tin oxide (FTO) conductive glass was cut into dimensions of 2 cm × 1 cm and wrapped with aluminum foil to make a deposition area of 1 cm × 1 cm and then placed in a tube furnace with a set temperature of 480 °C. The precursor solution was added to the injection pump and dispersed into droplets by using an ultrasonic atomizer. During the experiment, 40 mL of precursor solution was injected at a speed of 1.6 mL min$^{-1}$, which equally matched the power of the ultrasonic atomizer. Using air as the carrier gas, the precursor was fed into a tubular furnace. After deposition, the film was annealed in a muffle furnace at 600 °C for 3 h at a heating rate of 10 °C min$^{-1}$. The Sn/$\beta$-Fe$_2$O$_3$ films were prepared using the same spray pyrolysis method by adding a certain amount of tetrabutyltin (C$_{16}$H$_{36}$Sn, analytical reagent, Aladdin) ethanol solution to the precursor solution so that the Sn atom concentration accounted for 1%, 2%, 3%, and 4% of the total Sn and Fe atoms. The CoFe-LDH @ Sn/$\beta$-Fe$_2$O$_3$ photoanodes were prepared by a hydrothermal method. The as-prepared Sn/$\beta$-Fe$_2$O$_3$ photoanodes were put into a 100 mL hydrothermal kettle, 50 mL of a solution containing 0.002 mol L$^{-1}$ cobalt nitrate hexahydrate (Co(NO$_3$)$_2$·6H$_2$O, Sinopharm Chemical Reagent), 0.002 mol L$^{-1}$ iron(III) nitrate nonahydrate (Fe(NO$_3$)$_3$·9H$_2$O, analytical reagent, Aladdin)), 0.005 mol L$^{-1}$ urea (Aladdin) and 0.001 mol L$^{-1}$ trisodium citrate was added, and the reaction was carried out in an oven at 120 °C for 5 h.

### Characterization

To identify the crystal structures of the $\beta$-Fe$_2$O$_3$ photoanodes, they were measured by powder X-ray diffraction (XRD, Rigaku Ultima III, Cu Kα radiation, $\lambda = 1.54178$ Å) at 40 kV and 40 mA. The surface morphology of the $\beta$-Fe$_2$O$_3$ photoanodes was examined by a high-resolution scanning electron microscope (HRSEM, ZEISS ULTRA 55 at an accelerating voltage of 5 kV). Raman spectra of $\beta$-Fe$_2$O$_3$ photoanodes were characterized with a confocal laser Raman spectrometer (Japan, Horiba, LabRAM Aramis). X-ray photoemission spectroscopy (XPS, PHI 5000 VersaProbe) was used to characterize the content and valence of Sn, O, Fe, and Co, and the binding energy was calibrated by the adventitious carbon C 1$s$ line at 284.8 eV. The optical absorption

spectra of the photoanode were tested on a UV–Visible–NIR (near-infrared) spectrophotometer (PerkinElmer, UV3600 UV–Vis–NIR spectrophotometer). Transmission electron microscopy (TEM) and high-resolution transmission electron microscopy (HRTEM) images were obtained on an FEI Tecnai G2 F30. High-angle annular dark field (HAADF) scanning transmission electron microscopy (STEM) images were obtained by a JEOL JEM-ARM200F microscope incorporated with a spherical aberration correction system for STEM. The concentrations of cations in the condensates were examined by ICP-OES (PerkinElmer Instruments, PTIMA 5300 DV).

## PEC measurements

The PEC measurements were carried out in a PEC cell with an electrochemical analyser (CHI-760E, CH Instrument, Shanghai) in a three-electrode system including a reference electrode consisting of Ag/AgCl placed in a saturated KCl solution, Pt foil as the counter electrode, and $\beta$-$Fe_2O_3$ photoanodes as working electrodes. The electrolyte was a 1 M KOH aqueous solution for freshwater and 1 M KOH with 0.5 M NaCl for simulated seawater. The potential was reported vs. the reversible hydrogen electrode (RHE) with $E_{RHE} = E_{Ag/AgCl} + 0.197 + 0.0591$ pH[38–41]. The photocurrent density was measured under AM 1.5 G light source, and the light intensity was 100 mW cm$^{-2}$. A Newport 91150 V standard silicon cell was used as the reference standard for calibration. Mott-Schottky analysis was performed at bias potentials from 0.5 to 1.5 V vs. RHE[42,43]. AC electrochemical impedance was obtained at a bias of 1.6 $V_{RHE}$ over the frequency range of 100 kHz to 1 Hz[44,45]. The PEC stabilities were tested at a constant potential of 1.6 $V_{RHE}$ under LED-simulated sunlight sources through illumination from the front side. No iR compensation was used in any electrochemical test. All tests were conducted at room temperature (~25 °C) and in an air environment.

In the PEC test of the H/D kinetic isotope effect, the electrolyte was measured with a pH meter to keep the concentrations of $OH^-$ and $OD^-$ in the solution the same (pD = $pH_{read}$ + 0.4). $D_2O$ was purchased from Bide Pharmatech Ltd. (99.9% atom %D). The pD values were adjusted by NaOD (Aladdin, 30 wt% solution in $D_2O$, 99.5%). In the current time curve, the photocurrent density value after 50 s of reaction was selected as the steady-state value for the calculation of $j_{H2O}/j_{D2O}$ and $j_{Sn}/j_{Pure}$ (Supplementary Fig. 11).

All photoanodes after the reaction were removed from the electrolyte and washed with flowing deionized water for 20 s to remove the residual electrolyte on the surface. Then, the cleaned photoanodes were further characterized and analysed.

## Time-of-flight secondary ion mass spectrometry (TOF-SIMS) tests

TOF-SIMS tests were carried out by PHI nanoTOF II Time-of-Flight SIMS. $Bi_3^{++}$ with an energy of 30 eV was used in the acquisition phase in high mass resolution mode. An Ar ion gun with an energy of 4 kV was used in the sputter phase with a sputter rate of 0.4 nm s$^{-1}$ on $SiO_2$. Before the $\beta$-$Fe_2O_3$ photoanode was tested, the reactions in the electrolyte with 1 M KOH + 0.5 M NaCl and 20 wt% $H_2^{18}O$ for 100 h were carried out.

## X-ray absorption near-edge structure (XANES) tests

Soft X-ray absorption near-edge structure (XANES) measurements were performed at the Beijing Synchrotron Radiation Facility (BSRF), 4B9B beamline. The O-K edge and Fe-L edge spectra were collected in total electron yield (TEY) mode by measuring the sample current with an amperemeter. All spectra were normalized to the intensity of the incident beam (I0), which was measured simultaneously with the current emitted from a gold mesh located after the last optical elements of the beamline. The photon energy was calibrated using the Au-4$f$ core level at 84.0 eV in binding energy by measuring a clean polycrystalline gold foil that is electrically connected to the sample.

## Computational processing

The calculations on pure and Sn/$\beta$-$Fe_2O_3$ were implemented in the VASP (Vienna Ab initio Simulation Package) based on density functional theory, with a projected-augmented-wave method in the scheme of generalized-gradient approximation. The strong on-site Coulomb repulsion among the localized Fe 3$d$ electrons was described with the generalized-gradient approximation + U approach (U is the strength of the on-site Coulomb interaction). The exchange-correlation effects were treated using the generalized-gradient approximation (GGA) in the Perdew-Burke-Ernzerhof parametrization, with spin-polarized effects considered. The calculated unit cell contains 96 Fe atoms and 144 oxygen atoms. In doping calculations, different numbers of Fe atoms were replaced with Sn atoms. The replacement site is calculated as the position with the lowest energy.

## Data availability

All data are available in the main text and Supplementary Information. The data generated in this study are provided in the Source data file. Source data are provided with this paper.

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

## Acknowledgements

The authors thank the National Science Fund for Distinguished Young Scholars (No. 22025202, Z.L.), National Key Research and Development Program of China (Nos. 2018YFA0209303, Z.L. and 2021YFA1502100, J.F.), National Natural Science Foundation of China (No. 51972165, Z.L.) and Natural Science Foundation of Jiangsu Province of China (No. BK20202003, Z.Z.) for financial support. We are indebted to Prof. Yixin Zhao (Shanghai Jiaotong University) for discussions.

## Author contributions

Z.L. constructed the concept and designed the project. Z.L. supervised the study. N.Z., Y.L., J.F., W.W., C.L., J.W., W.H., and Z.Z. advised on the research. C.H.L. and R.F. collected and analysed the experimental data. C.H.L. and Z.L. wrote the manuscript. Z.L. and J.F. revised the manuscript. All the authors contributed to the discussions about the manuscript.

## Competing interests

The authors declare no competing interests.
