## [Peer Review File · Nature Communications]

Long-term durability of metastable β -Fe₂O₃ photoanodes in highly corrosive seawaterREVIEWER COMMENTS

Reviewer #1 (Remarks to the Author):

The article “Ultradurability of metastable β -Fe₂O₃ photoanodes in highly corrosive seawater” reports an important effect of tin when used to dop β -Fe₂O₃ photoanodes, rendering them more stable and energy efficient – higher current density.

The manuscript is adequate for being published at Nature Communications after addressing the comments indicated below.

Some comments:

Line 42 – “a theoretical solar-to-hydrogen efficiency of 20.9%” – what theory do the authors refer to? This expression is often used but it is also often meaningless, since it is not the thermodynamic efficiency. Please review and correct it.

Line 43 – please review references 6 and 7 since they seem to be not appropriate;

Lines 68-70 – “The saturated photocurrent density of the 2% Sn/ β -Fe₂O₃ photoanodes reaches 2.21 mA cm⁻² at 1.6 VRHE, which is 8.5 times that of β -Fe₂O₃ photoanodes (Supplementary Fig. 2a).” – normally the current density is obtained either at the on-set of the dark current (which is not indicated) or at the thermodynamic electrolysis potential – 1.48 V @ 25 °C.

Lines 71-73 – “...while the PEC performance improvement is partly due to the increased carrier concentration (Supplementary Fig. 3)” – please clarify this statement.

Figure S2 – where is the on-set of the dark current? Why the current density increases from 4 -> 1 -> 3 -> 2 and the impedance goes from 2 -> 1 -> 3 -> 4?

The onset of the photocurrent is quite shifted to the higher potentials, at ca. 1.15 VRHE. This means that the generated photopower is quite small. Please comment on that. Compared with for example <http://dx.doi.org/10.1016/j.nanoen.2017.05.051>, where the onset is at ca. 0.5 VRHE.

Lines 181-182 – “The advantages of high valence cation-doped Sn dispersed in the bulk phase in increasing the carrier concentration and conductivity can also be fully demonstrated.” – the oxidation state of the tin can be +4 or -4; which carrier you mean? electrons or holes. Please elaborate more the discussion.

Reviewer #2 (Remarks to the Author):

Changhao Liu et al's manuscript presents a photoanode that is Sn doped β -Fe₂O₃ and is able to perform photoelectrocatalytic seawater splitting for 1440 hours without significant instability issues. The stability of PEC catalysts in seawater is a crucial problem that presents a significant challenge to realizing their practical application. Even though the activity is not exceptional, the authors' work highlights the potential of low-cost materials like Fe₂O₃ in achieving long-term stable PEC seawater splitting, which is a promising development. Additionally, the manuscript presents strong evidence of the stable OER of Sn/ β -Fe₂O₃ through the use of X-ray absorption near-edge structure and time-of-flight secondary ion mass spectrometry. However, before it can be published in Nature Communications, a few questions must be addressed.

1. The authors are encouraged to provide an explanation for the over 20% increase in current observed in Fig. 2d during the first 700 hours.

2. It is necessary for the authors to clarify the experimental conditions under which the ICP samples were collected, which can be found in Supplementary Table 2.
3. The Raman spectrum presented in Fig. 3d may be perplexing to readers, and it would be helpful if the authors explained the reason for annealing the samples and why they preferred to use Raman instead of XRD to characterize the materials, which may produce clearer results. Additionally, it would be interesting to know if the samples were adequately washed before characterization, as even a small amount of residual seawater could contain a significant amount of NaCl.
4. In line 116, Fig. 3g should be corrected to Fig. 3c, while Fig. 3c in line 120 should be corrected to Fig. 3d.
5. It would be beneficial if the authors could measure the conductivities of β -Fe₂O₃ and Sn/ β -Fe₂O₃, as they claim that the addition of Sn can enhance the conductivity of β -Fe₂O₃.
6. Deeper XPS analysis for Sn should be provided, as the authors suggest that uniformly dispersed Sn in the bulk phase can help stabilize β -Fe₂O₃. However, since the thickness of the Sn/ β -Fe₂O₃ layer is over 200 nm, the 10 nm measurement is likely surface-based.

Reviewer #3 (Remarks to the Author):

Photoelectrochemical (PEC) water splitting using Earth-abundant seawater and sunlight is a promising way to produce green hydrogen on a large scale. However, this is still challenging due to the high corrosiveness of seawater, particularly the presence of high-concentration Cl⁻ in seawater. In this paper, Liu and co-workers report a Sn-doped β -Fe₂O₃ photoanode with exceptional stability for PEC seawater oxidation. Sn dopant was found to enhance the metal-oxygen bonding energy in β -Fe₂O₃ and hinder the transfer of protons to the lattice oxygen, thereby inhibiting excessive surface hydration and Cl⁻ coordination. As a result, a record durability of 1440 h was achieved with the Sn/ β -Fe₂O₃ photoanode without any surface modification.

Overall, the results presented here are intriguing and the manuscript is concisely and clearly written. I have only some minor questions regarding this paper:

- 1) As pointed out by the authors, the Sn dopants also slowly leached out of the β -Fe₂O₃ photoanode due to lattice reconstruction after long-term operation (1000 h). Could the author comment on if the self-healing or dynamic stability concept (as proposed in Refs. 20-21 for Fe-based OER catalysts) can be brought to this system to stabilize the Sn dopant? Can this lead to the ultimate stability of the β -Fe₂O₃ photoanode?
- 2) Although the size (1 × 1 cm) of the photoanode is reported in the Methods, it would be more straightforward to report photocurrent density instead of photocurrent in Figure 2a and 2d.
- 3) If possible, the current-potential curves of the photoanodes after the stability tests are suggested to be provided, as they provide meaningful information on the change of the onset potentials after the stability test.
- 4) Line 43, Refs 6 and 7 are not about β -Fe₂O₃, please double-check the accuracy of the reference list.

Point-by-point responses for Nature Communications manuscript

(ID: NCOMMS-22- 49861-T)

Manuscript Type: Article

Title: Ultradurability of metastable β -Fe₂O₃ photoanodes in highly corrosive seawater.

Author(s): Changhao Liu, Ningsi Zhang, Yang Li, Rongli Fan, Wenjing Wang, Jianyong Feng, Chen Liu, Jiaou Wang, Weichang Hao, Zhaosheng Li, Zhigang Zou

General response:

We sincerely thank the editor, editorial staff and all reviewers for their critical comments that we have based on to improve the quality of our manuscript. The manuscript has been modified point-by-point after addressing all the suggestions as listed below.

(Our response is given in blue and the corrections in the revised manuscript are shown in red)

Point-by-point responses to Reviewer(s)

Reviewer #1:

The article “Ultra-durability of metastable β -Fe₂O₃ photoanodes in highly corrosive seawater” reports an important effect of tin when used to dop β -Fe₂O₃ photoanodes, rendering them more stable and energy efficient – higher current density.

The manuscript is adequate for being published at *Nature Communications* after addressing the comments indicated below.

Response:

We are very grateful to the reviewer. The review comments are of great significance to further improve the manuscript. We have addressed the comments point-by-point and made the corresponding changes accordingly in the revised manuscript.

Some comments:

1. Line 42 – “a theoretical solar-to-hydrogen efficiency of 20.9%” – what theory do the authors refer to? This expression is often used but it is also often meaningless, since it is not the thermodynamic efficiency. Please review and correct it.

Response:

We thank the reviewer for the insightful comments. The statement here is not proper. The theoretical

solar-to-hydrogen efficiency of 20.9% described here is simply calculated based on the optical absorption range corresponding to the band gap of 1.9 eV of β -Fe₂O₃, combined with the AM 1.5 G solar spectrum^{R1}. However, a series of factors, such as specific band structure, transition model of electrons, carrier relaxation and recombination, optical absorption loss, and resistance loss, are not considered here. These processes will inevitably occur and have a great impact. Our original intention here is only to emphasize that β -Fe₂O₃ has a narrower band gap and a wider optical absorption range than α -Fe₂O₃.

[r1] Li, Z. S. et al. Photoelectrochemical cells for solar hydrogen production: current state of promising photoelectrodes, methods to improve their properties, and outlook. *Energy Environ. Sci.* **6**, 347–370, (2013).

Therefore, the corresponding description in the manuscript is modified as follows.

Lines 41–45 (Original manuscript lines 41–43):

Recently, β -Fe₂O₃ entered our research field as a metastable phase of iron oxide. Because of its narrower band gap (1.9 eV) compared with α -Fe₂O₃ (2.1 eV), the theoretical optical absorption band edge can be extended to approximately 650 nm. Thus, it has a higher theoretical solar-to-hydrogen efficiency than α -Fe₂O₃. At the same time, β -Fe₂O₃ also shows good stability in photoelectrochemical alkaline water splitting^{20, 21}.

2. Line 43 – please review references 6 and 7 since they seem to be not appropriate;

Response:

Thanks for the reviewer’s comment. Here is a citation error. We have added the correct and corresponding references here, and other corresponding reference numbers have also been changed.

References

Lines 325–329 (Original manuscript lines 298):

[20] Zhang, N. S. et al. Paving the road toward the use of β -Fe₂O₃ in solar water splitting: Raman identification, phase transformation and strategies for phase stabilization. *Natl. Sci. Rev.* **7**, 1059–1067, (2020).

[21] Li, Y. et al. Metastable-phase β -Fe₂O₃ photoanodes for solar water splitting with durability exceeding 100 h. *Chinese J. Catal.* **42**, 1992–1998, (2021).

3. Lines 68-70 – “The saturated photocurrent density of the 2% Sn/ β -Fe₂O₃ photoanodes reaches 2.21 mA cm⁻² at 1.6 V_{RHE}, which is 8.5 times that of β -Fe₂O₃ photoanodes (Supplementary Fig. 2a).” – normally the current density is obtained either at the on-set of the dark current (which is not indicated) or at the thermodynamic electrolysis potential – 1.48 V @ 25 °C.

Response:

We are greatly grateful to the reviewer for the nice question, which can help us revise the discussion

in the manuscript more accurately. The photoelectric current at 1.6 V_{RHE} is selected for comparison, considering the following two factors. i. The photocurrent of the common α -Fe₂O₃ photoanode has reached saturation at 1.6 V_{RHE}, so we choose this potential to compare. Since photogenerated carriers have reached saturation, the photocurrent will not increase with the increase of bias voltage. ii. In the supplementary dark current test, it is found that the on-set of the dark current of β -Fe₂O₃ photoanodes is usually at 1.7~1.8 V_{RHE}.

Fig. R1. Dark current density of 0% to 4% Sn/ β -Fe₂O₃ photoanodes.

During the long-term stability test, we found that the on-set of the dark current would shift negatively due to the formation of FeOOH on the surface in situ as electrocatalysts. However, the on-set of the dark current is always after 1.6 V_{RHE}, so it can be determined that the current at 1.6 V_{RHE} is contributed by the photocurrent. Considering all β -Fe₂O₃ with different Sn concentrations and the photoanodes before and after the stability tests, we choose the photocurrent density at 1.6 V_{RHE} for comparison. And the subsequent stability test is also conducted under this bias voltage.

We modified Supplementary Fig. 2 and added the dark current of 0% to 4% Sn/ β -Fe₂O₃ photoanodes as follows.

Supplementary Fig. 2. Photoelectrochemical tests of β -Fe₂O₃ with different Sn concentrations.

a, Photocurrent density of 0% to 4% Sn/ β -Fe₂O₃ photoanodes at 0.6–1.7 V_{RHE} in 1 M KOH + 0.5 M NaCl simulated seawater under one sun illumination. b, Dark current density of 0% to 4% Sn/ β -Fe₂O₃ photoanodes.

The corresponding description in the manuscript is modified as follows.

Lines 70–72 (Original manuscript lines 68–70):

The photocurrent density of the 2% Sn/ β -Fe₂O₃ photoanodes reaches 2.21 mA cm⁻² at 1.6 V_{RHE},

which is 8.5 times that of $\beta\text{-Fe}_2\text{O}_3$ photoanodes (Supplementary Fig. 2).

4. Lines 71-73 – “...while the PEC performance improvement is partly due to the increased carrier concentration (Supplementary Fig. 3)” – please clarify this statement.

Response:

Thank the reviewer for the constructive suggestions. In the original manuscript, we do not have a detailed description and analysis of the increased carrier concentration in Sn/ $\beta\text{-Fe}_2\text{O}_3$. The carrier concentrations in $\beta\text{-Fe}_2\text{O}_3$ and Sn/ $\beta\text{-Fe}_2\text{O}_3$ can be calculated by the slope of the tangent of the M-S curve. With the increase of carrier concentration, the conductivity of Sn/ $\beta\text{-Fe}_2\text{O}_3$ is also improved, which is consistent with the AC electrochemical impedance spectra. We have supplemented the detailed discussion in Supplementary Fig. 4 and revised our manuscript.

We have added relevant discussion in Supplementary Fig. 4.

Supplementary Fig. 4. Carrier concentration and AC impedance of $\beta\text{-Fe}_2\text{O}_3$ and Sn/ $\beta\text{-Fe}_2\text{O}_3$.

a, Mott-Schottky plots of $\beta\text{-Fe}_2\text{O}_3$ and Sn/ $\beta\text{-Fe}_2\text{O}_3$ photoanodes measured at 1.6 V_{RHE} . b, High-frequency part of AC electrochemical impedance spectra of $\beta\text{-Fe}_2\text{O}_3$ and Sn/ $\beta\text{-Fe}_2\text{O}_3$ photoanodes.

We used Mott–Schottky relationship to determine the donor concentration (N_D):

$$\frac{1}{C_{SC}^2} = \frac{2}{q\epsilon\epsilon_0 N_D} \left(V - V_{fb} - \frac{kT}{q} \right)$$

where C_{SC} is the space charge capacitance, q is the elementary charge, ϵ_0 is the permittivity of free space, and ϵ is the dielectric constant of $\beta\text{-Fe}_2\text{O}_3$ ³⁷. The slope of the tangent in Supplementary Fig. 4a is inversely proportional to the carrier concentration:

$$Slope = \frac{2}{e\epsilon\epsilon_0 N_D}$$

where e is the electron charge. Thus, it can be estimated that the carrier (electrons) concentration of Sn/ $\beta\text{-Fe}_2\text{O}_3$ is 8.4 times that of $\beta\text{-Fe}_2\text{O}_3$.

In Supplementary Fig. 4b, the half-circle fitted by the AC electrochemical impedance spectra in the high frequency region is related to R3/CPE2 in the equivalent circuit model, which is assigned to the electron transport inside the electrode. The R3 values of $\beta\text{-Fe}_2\text{O}_3$ and Sn/ $\beta\text{-Fe}_2\text{O}_3$ calculated

from the fitting results are 1065.0 Ω and 299.8 Ω respectively, which means that the conductivity of Sn/ β -Fe₂O₃ is much higher than that of β -Fe₂O₃.

References

Lines 369–371 (Original manuscript lines 343):

[37] Franking, R. *et al.* Facile post-growth doping of nanostructured hematite photoanodes for enhanced photoelectrochemical water oxidation. *Energy Environ. Sci.* **6**, 500–512, (2013).

The corresponding description in the manuscript is modified as follows.

Lines 72–80 (Original manuscript lines 70–74):

The Sn dopants do not affect the light absorption of the β -Fe₂O₃ photoanodes, while the PEC performance improvement is partly due to the increased electron concentration caused by the doping of high-valence cation Sn⁴⁺. The carrier concentration of Sn/ β -Fe₂O₃ is approximately 8.4 times that of β -Fe₂O₃, and correspondingly, its bulk conductivity is also improved, according to Mott-Schottky plots and AC electrochemical impedance spectra in the low-frequency region. Sn can simultaneously adjust the chemical field at the semiconductor/electrolyte interface, which significantly reduces the AC impedance of the interface transfer kinetics (Supplementary Fig. 3 and 4).

5. Figure S2 – where is the on-set of the dark current? Why the current density increases from 4 -> 1 -> 3 -> 2 and the impedance goes from 2 -> 1 -> 3 -> 4?

The onset of the photocurrent is quite shifted to the higher potentials, at ca. 1.15 V_{RHE}. This means that the generated photo-power is quite small. Please comment on that. Compared with for example <http://dx.doi.org/10.1016/j.nanoen.2017.05.051>, where the onset is at ca. 0.5 V_{RHE}.

Response:

Thanks for the reviewer's valuable comments.

Fig. R2. Dark current density of 0% to 4% Sn/ β -Fe₂O₃ photoanodes.

The on-set of the dark current is between 1.7~1.8 V_{RHE}. It is related to the electrocatalytic properties of the surface of photoanodes. The AC electrochemical impedance spectra in the low frequency region in Supplementary Fig. 3b can reflect the transfer kinetics at the interface, which has a strong correlation with the change of doping concentration and the on-set of the dark current.

We modified Supplementary Fig. 2 and added the dark current of 0% to 4% Sn/ β -Fe₂O₃ photoanodes as follows.

Supplementary Fig. 2. Photoelectrochemical tests of β -Fe₂O₃ with different Sn concentrations. a, Photocurrent density of 0% to 4% Sn/ β -Fe₂O₃ photoanodes at 0.6–1.7 V_{RHE} in 1 M KOH + 0.5 M NaCl simulated seawater under one sun illumination. b, Dark current density of 0% to 4% Sn/ β -Fe₂O₃ photoanodes.

In addition to interface transfer kinetics, the photocurrent density is also related to the separation and transmission of carriers in the bulk. The low-frequency impedance shows a trend of 2 \rightarrow 1 \rightarrow 3 \rightarrow 4. This is because the addition of a small amount of Sn improves the electronic environment of the surface and is conducive to charge transfer and matter exchange. When a large amount of Sn accumulates on the surface, Fe as the reactive active center was blocked, so the impedance increased. Another important factor affecting the photocurrent density is the conductivity of the semiconductor. The Fermi level of the semiconductor will be closer to the conduction band with the increase of doping concentration, and the carrier concentration and conductivity increased. The interface reaction rate and the bulk transfer of carriers jointly determine that the photocurrent density goes from 4 \rightarrow 1 \rightarrow 3 \rightarrow 2.

The photocurrent on-set potential of β -Fe₂O₃ is quite high here, which is due to the absence of any electrocatalyst or passivation layer on the photoanodes surface. A large number of surface trapped states will have serious effects. But in this work, we only discuss the activity and stability of bare β -Fe₂O₃ photoanodes. Besides, the adding of Sn will bring a negative effect, that is, it will introduce more defect levels in the band gap. Photogenerated electrons and holes are trapped in the defect levels, and the charge recombination will also be more serious. More surface trapped states will also bind the carriers, resulting in the reduction of the photovoltage. For this reason, the on-set potential will shift to the higher potential with the increase of Sn concentration. Besides, the steady-state photovoltage test is carried out (Fig. R3). The photovoltage of Sn-doped photoanode reduced, which means that the generated photo-power became smaller.

Fig. R3. Steady-state photovoltage of $\beta\text{-Fe}_2\text{O}_3$ and Sn/ $\beta\text{-Fe}_2\text{O}_3$ photoanodes.

6. Lines 181-182 – “The advantages of high valence cation-doped Sn dispersed in the bulk phase in increasing the carrier concentration and conductivity can also be fully demonstrated.” – the oxidation state of the tin can be +4 or -4; which carrier you mean? electrons or holes. Please elaborate more the discussion.

Response:

Thank you for raising this valuable question. The Sn mentioned here is a +4-valence cation, and the high valence cation doping can provide more electrons as a donor. For n-type semiconductors, more cation doping means that the Fermi level is closer to the conduction band, and the conductivity will be improved.

The corresponding description in the manuscript is modified as follows, and the role of high-valent cations is discussed more.

Lines 72–74 (Original manuscript lines 70–72):

The Sn dopants do not affect the light absorption of the $\beta\text{-Fe}_2\text{O}_3$ photoanodes, while the PEC performance improvement is partly due to the increased electron concentration caused by the doping of high-valence cation Sn^{4+} .

Lines 189–190 (Original manuscript lines 180–182):

The advantages of donor Sn^{4+} dispersed in bulk phase in improving electron concentration and conductivity can also be fully demonstrated.

Reviewer #2:

Changhao Liu et al's manuscript presents a photoanode that is Sn doped β -Fe₂O₃ and is able to perform photoelectrocatalytic seawater splitting for 1440 hours without significant instability issues. The stability of PEC catalysts in seawater is a crucial problem that presents a significant challenge to realizing their practical application. Even though the activity is not exceptional, the authors' work highlights the potential of low-cost materials like Fe₂O₃ in achieving long-term stable PEC seawater splitting, which is a promising development. Additionally, the manuscript presents strong evidence of the stable OER of Sn/ β -Fe₂O₃ through the use of X-ray absorption near-edge structure and time-of-flight secondary ion mass spectrometry. However, before it can be published in *Nature Communications*, a few questions must be addressed.

Response:

We sincerely thank the reviewer's comments. The proposed suggestions are valuable and helpful for improving our work. We have carefully revised the manuscript and replied to the comments point-by-point shown below.

1. The authors are encouraged to provide an explanation for the over 20% increase in current observed in Fig. 2d during the first 700 hours.

Response:

We thank the reviewer for the insightful comments. In the whole process of the stability test, the change of photocurrent density shows a trend of slowly increasing at first, then gradually decreasing, and finally stabilizing. This is because at the initial stage of the reaction, β -Fe₂O₃ on the surface is converted into FeOOH, which XPS, Raman and other experiments can also prove (Fig. 3). FeOOH is an efficient electrocatalyst, so the photocurrent density increased at the early stage of the reaction. Because of the infiltration corrosion of Cl⁻ along with the excessive surface reconstruction and the loss of Sn, the photocurrent density had a downward trend after 700 h. However, due to the relatively uniform distribution of Sn in the whole film, the reconstruction and corrosion are controlled within a certain range of the Sn/ β -Fe₂O₃ surface. The evolution of the surface has reached a relatively stable state after a period of time, so the photocurrent density tends to be stable after 1000 h.

In the past period, we have extended the stability test to 3000 h (Fig. 3d). In the later period of the stability test, the photocurrent is kept in a relatively stable range.

The following is the replacement of the stability image in the manuscript, adding 3000 h stability data.

Fig. 2. | PEC properties of the β -Fe₂O₃ photoanode in simulated seawater.

a, Stability test of β -Fe₂O₃ (in 1 M KOH, 1 M KOH/0.5 M NaCl, 1 M KOH/saturated NaCl solution) and Sn/ β -Fe₂O₃ (in 1 M KOH/saturated NaCl solution) for 100 h. b, AC electrochemical impedance spectra of Sn/ β -Fe₂O₃ before and after the reaction. c, HAADF images of the Sn/ β -Fe₂O₃ photoanode after 100 h of reaction in 1 M KOH/0.5 M NaCl. d, Stability test of Sn/ β -Fe₂O₃ in 1 M KOH/0.5 M NaCl for 3000 h. e, Summary of the photoanode stability of PEC (simulated) seawater splitting over the years. Detailed information can be found in Supplementary Table 1.

The following is the description added to the text.

Lines 201–217 (Original manuscript lines 192):

Such characteristics make the stability rise continuously at first, then decline slowly, and finally maintain a stable range of fluctuations. At the initial stage of the reaction, the β -Fe₂O₃ on the surface is converted into FeOOH in situ (Fig. 3). FeOOH itself is an efficient electrocatalyst, so the photocurrent increased. Because of the infiltration corrosion of Cl⁻ along with the excessive surface reconstruction and the loss of Sn, the photocurrent density had a downwards trend after 700 h. It can be analyzed from Supplementary Fig. 10 that due to the electrocatalytic effect of surface FeOOH after 1000-h reaction, the on-set potential moved to the negative direction by approximately 0.1 V_{RHE}. However, the photocurrent density at 1.6 V_{RHE} was reduced with the influence of surface reconstruction and corrosion. Owing to the relatively uniform distribution of Sn in the whole film, reconstruction and corrosion were controlled within a certain range of β -Fe₂O₃ surface, instead of continuing to the deep layer. In addition, while the surface reconstruction was accompanied by the loss of Sn, there would also be a sedimentation equilibrium on the surface, and deposition would occur, which forms a dynamic balance of corrosion, metal loss, deposition and protection. A dynamic stable state of surface evolution was established after a period of time, so the photocurrent density tended to be stable after 1000 h.

2. It is necessary for the authors to clarify the experimental conditions under which the ICP samples were collected, which can be found in Supplementary Table 2.

Response: We appreciate the reviewer's comments very much. We have supplemented the relevant information of the ICP test in the method and Supplementary Materials.

The instrument information of ICP is supplemented in the Methods section.

Lines 437–438 (Original manuscript lines 402):

The concentrations of cations in condensates were examined by ICP-OES (PerkinElmer Instruments, PTIMA 5300 DV).

We have added the test conditions of ICP experiment in Supplementary Table 2 as follows.

Samples	Fe ion concentration in electrolyte (mg/L)
β -Fe ₂ O ₃	0.071
Sn/ β -Fe ₂ O ₃	0.036

Supplementary Table 2. ICP measurement of Fe ions in the electrolyte after the reaction.

Take the alkaline simulated seawater electrolyte after 100 h of reaction, and measure the content of Fe ion in the solution. Based on this, we can analyse the loss of Fe caused by surface reconstruction in the stability test. Dilution and pH adjustment of the same multiple was conducted before the test.

3. The Raman spectrum presented in Fig. 3d may be perplexing to readers, and it would be helpful if the authors explained the reason for annealing the samples and why they preferred to use Raman instead of XRD to characterize the materials, which may produce clearer results. Additionally, it would be interesting to know if the samples were adequately washed before characterization, as even a small amount of residual seawater could contain a significant amount of NaCl.

Response: We sincerely thank the reviewer's comments. The annealing treatment further confirms the hydration and reconstruction of the surface of β -Fe₂O₃ after a long-term reaction in the electrolyte. β -Fe₂O₃ was transformed into FeOOH during the reconstruction, and FeOOH would be transformed into stable phase α -Fe₂O₃ after annealing. Through such treatment, the process of surface reconstruction and material transformation can be more intuitively displayed.

In our tests, we found that the signal-to-noise ratio of the XRD of the β -Fe₂O₃ layer grown on the surface of the FTO conductive glass was weak. Because the reconstruction during the reaction only occurred on the surface of the film, no more valuable signal can be detected in the XRD test. The confocal Raman can detect the signal of trace substances on the sample surface, so the surface reconstruction can be well detected here.

The corresponding description in the manuscript is modified as follows.

Lines 125–133 (original manuscript lines 119–124):

Confocal Raman spectroscopy is used to observe the speciation evolution of trace substances on the surface of photoanode before and after the reaction. After 100 h and 1000 h of seawater splitting, there are two peaks of M–OOH at approximately 470 cm⁻¹ and 550 cm⁻¹ ^{24,25}, which echo the change

in the O 1 s XPS peak. FeOOH, which is hydrated and reconstituted during the reaction, is also more prone to dehydration and sintering at high temperatures. Thus, the α -Fe₂O₃ peak can be observed when the β -Fe₂O₃ photoanodes are further calcined at 600°C.

The specific steps of cleaning are as follows: remove the photoanodes after reaction for a period of time from the electrolyte, and immediately wash them with flowing deionized water for 20 s. Make sure there is no residual electrolyte. If Cl⁻ or NaCl is detected later, it is considered that they are strong adsorption ions formed on the surface of β -Fe₂O₃, or ions participating in the surface reconstruction and entering the lattice. In addition, no NaCl signal was detected for Sn/ β -Fe₂O₃ photoanode after 100-h reaction which was treated with the same cleaning method. This is because Cl⁻ in the electrolyte has not yet formed strong adsorption or participated in surface reconstruction. This comparison also shows that the cleaning of the photoanode after the reaction is effective.

In the Methods section, we added relevant descriptions.

Lines 462–464 (original manuscript lines 425):

All photoanodes after reaction were taken out of the electrolyte and washed with flowing deionized water for 20 s to remove the residual electrolyte on the surface. Then the cleaned photoanodes were further characterized and analyzed.

4. In line 116, Fig. 3g should be corrected to Fig. 3c, while Fig. 3c in line 120 should be corrected to Fig. 3d.

Response: We sincerely thank the reviewer's correction.

The corresponding description in the manuscript is modified as follows.

Lines 121–123 (original manuscript lines 115–117):

The XPS peak of the Sn 3d signal disappeared after 1000 h of reaction (Fig. 3c), indicating that Sn is also slowly lost during lattice reconstruction by surface hydration.

Lines 125–129 (original manuscript lines 119–122):

Confocal Raman spectroscopy is used to observe the speciation evolution of trace substances on the surface of photoanode before and after reaction (Fig. 3d). After 100 h and 1000 h of seawater splitting, there are two peaks of M–OOH at approximately 470 cm⁻¹ and 550 cm⁻¹^{26,27}, which echo the change in the O 1 s XPS peak.

5. It would be beneficial if the authors could measure the conductivities of β -Fe₂O₃ and Sn/ β -Fe₂O₃, as they claim that the addition of Sn can enhance the conductivity of β -Fe₂O₃.

Response: We are very grateful to the reviewers for the suggestions. We compare the conductivities of β -Fe₂O₃ and Sn/ β -Fe₂O₃ by analysing the Nyquist impedance. The AC electrochemical impedance data are fitted into an equivalent circuit model including two RC (a sub-circuit containing a resistance and a capacitance in parallel) circuits, as shown in Fig R2.

(1) R1 is the solution resistance.

(2) R2/CPE1 in the low-frequency region with the larger resistance represents interface transfer

kinetics between n-type semiconductor electrode and electrolyte (Supplementary Fig. 3b).

(3) R3/CPE2 in the high-frequency region is assigned to the electron transport inside the electrode (Supplementary Fig. 4b).

By fitting the data, we obtained the corresponding parameter values of $\beta\text{-Fe}_2\text{O}_3$ and Sn/ $\beta\text{-Fe}_2\text{O}_3$ photoanodes in the equivalent circuit model, and the error was less than 2%.

Samples	R1 (Ω)	R2 (Ω)	R3 (Ω)	CPE1-T (F)	CPE1-P /	CPE2-T (F)	CPE2-P /
$\beta\text{-Fe}_2\text{O}_3$	17.61	148,290	1,065	1.3719×10^{-6}	0.90126	2.3622×10^{-6}	0.97877
Sn/ $\beta\text{-Fe}_2\text{O}_3$	14.93	19,137	299.8	3.5184×10^{-6}	0.89427	1.6896×10^{-5}	0.89833

Table R1. ICP measurement of Fe ions in the electrolyte after the reaction.

The R3 values of $\beta\text{-Fe}_2\text{O}_3$ and Sn/ $\beta\text{-Fe}_2\text{O}_3$ calculated from the fitting results are 1065.0 Ω and 299.8 Ω respectively, which means that the conductivity of Sn/ $\beta\text{-Fe}_2\text{O}_3$ is much higher than that of $\beta\text{-Fe}_2\text{O}_3$. Improved conductivity is due to the increase of carrier concentration in Sn/ $\beta\text{-Fe}_2\text{O}_3$, as discussed in Supplementary Fig. 4a.

$\beta\text{-Fe}_2\text{O}_3$ grown on the surface of conductive glass substrate has a certain orientation along the lattice direction of FTO, and its conductivity is anisotropic. The resistance perpendicular to the plane direction of the film measured by AC electrochemical impedance is more practical. At present, we cannot grow high-quality single-crystal $\beta\text{-Fe}_2\text{O}_3$ on other substrates and accurately measure its intrinsic conductivity.

We have added relevant discussion in Supplementary Fig. 4.

Supplementary Fig. 4. Carrier concentration and AC impedance of $\beta\text{-Fe}_2\text{O}_3$ and Sn/ $\beta\text{-Fe}_2\text{O}_3$.

a, Mott-Schottky plots of $\beta\text{-Fe}_2\text{O}_3$ and Sn/ $\beta\text{-Fe}_2\text{O}_3$ photoanodes measured at 1.6 V_{RHE} . b, High-frequency part of AC electrochemical impedance spectra of $\beta\text{-Fe}_2\text{O}_3$ and Sn/ $\beta\text{-Fe}_2\text{O}_3$ photoanodes.

We used Mott-Schottky relationship to determine the donor concentration (N_D):

$$\frac{1}{C_{SC}^2} = \frac{2}{q\epsilon\epsilon_0 N_D} \left(V - V_{fb} - \frac{kT}{q} \right)$$

where C_{SC} is the space charge capacitance, q is the elementary charge, ϵ_0 is the permittivity of free space, and ϵ is the dielectric constant of $\beta\text{-Fe}_2\text{O}_3$ ³⁷. The slope of the tangent in Supplementary Fig. 4a is inversely proportional to the carrier concentration:

$$Slope = \frac{2}{e\epsilon\epsilon_0 N_D}$$

where e is the electron charge. Thus, it can be estimated that the electron concentration of Sn/ β -Fe₂O₃ is 8.4 times that of β -Fe₂O₃.

In Supplementary Fig. 4b, the half-circle fitted by the AC electrochemical impedance spectra in high-frequency region is related to R3/CPE2 in the equivalent circuit model, which is assigned to the electron transport inside the electrode. The R3 values of β -Fe₂O₃ and Sn/ β -Fe₂O₃ calculated from the fitting results are 1065 Ω and 299.8 Ω respectively, which means that the conductivity of Sn/ β -Fe₂O₃ is much higher than that of β -Fe₂O₃.

References

Lines 369–371 (Original manuscript lines 343):

[37] Franking, R. *et al.* Facile post-growth doping of nanostructured hematite photoanodes for enhanced photoelectrochemical water oxidation. *Energy Environ. Sci.* **6**, 500–512, (2013).

The corresponding description of conductivity in the manuscript is modified as follows.

Lines 74–77 (original manuscript lines 70–72):

The electron concentration of Sn/ β -Fe₂O₃ is approximately 8.4 times that of β -Fe₂O₃, and correspondingly, its bulk conductivity is also improved, according to Mott-Schottky plots and AC electrochemical impedance spectra at low frequency region.

6. Deeper XPS analysis for Sn should be provided, as the authors suggest that uniformly dispersed Sn in the bulk phase can help stabilize β -Fe₂O₃. However, since the thickness of the Sn/ β -Fe₂O₃ layer is over 200 nm, the 10 nm measurement is likely surface-based.

Response: We appreciate the reviewer's valuable comments that help to improve the quality of our work. We conducted a deeper etching XPS experiment. To reflect the dispersion of bulk Sn, we increased the etching depth to 50 nm, 100 nm and 200 nm. Compared with the thickness of the film, this detection depth can better reflect the distribution of Sn in the bulk phase.

We added a new etching XPS experiment and replaced Supplementary Fig. 9.

Supplementary Fig. 9. Gradually varying Sn concentration in Sn/ β -Fe₂O₃.

Etching XPS of Sn 3d spectra of 2% Sn/ β -Fe₂O₃. Take the positions at depths of 0 nm, 50 nm, 100 nm and 200 nm from the surface layer for analysis.

Reviewer #3:

Photoelectrochemical (PEC) water splitting using Earth-abundant seawater and sunlight is a promising way to produce green hydrogen on a large scale. However, this is still challenging due to the high corrosiveness of seawater, particularly the presence of high-concentration Cl^- in seawater. In this paper, Liu and co-workers report a Sn-doped $\beta\text{-Fe}_2\text{O}_3$ photoanode with exceptional stability for PEC seawater oxidation. Sn dopant was found to enhance the metal-oxygen bonding energy in $\beta\text{-Fe}_2\text{O}_3$ and hinder the transfer of protons to the lattice oxygen, thereby inhibiting excessive surface hydration and Cl^- coordination. As a result, a record durability of 1440 h was achieved with the Sn/ $\beta\text{-Fe}_2\text{O}_3$ photoanode without any surface modification.

Overall, the results presented here are intriguing and the manuscript is concisely and clearly written. I have only some minor questions regarding this paper:

Response: We thank the reviewer for the very positive assessment of our work. The comments lead to further improvements in the quality of our work. According to the comments, we have modified our manuscript discussion and corresponding responses.

1) As pointed out by the authors, the Sn dopants also slowly leached out of the $\beta\text{-Fe}_2\text{O}_3$ photoanode due to lattice reconstruction after long-term operation (1000 h). Could the author comment on if the self-healing or dynamic stability concept (as proposed in Refs. 20-21 for Fe-based OER catalysts) can be brought to this system to stabilize the Sn dopant? Can this lead to the ultimate stability of the $\beta\text{-Fe}_2\text{O}_3$ photoanode?

Response: We appreciate the reviewer's valuable comments. Long-term stability test will lead to the slow leaching of Sn in the surface layer, resulting in a decline in photocurrent density after 700 h, as well as a certain degree of surface reconstruction and Cl^- erosion. In addition, we extended the stability test to 3000 h. We found that the subsequent photocurrent density did not decrease significantly, but remained relatively stable after the 1000-h reaction. Owing to the relatively uniform distribution of Sn in the whole film, reconstruction and corrosion were controlled within a certain range of $\beta\text{-Fe}_2\text{O}_3$ surface, instead of continuing to the deep layer. In addition, while the surface reconstruction was accompanied by the loss of Sn, there would also be a sedimentation equilibrium on the surface, and deposition would occur, which forms a dynamic balance of corrosion, metal loss, deposition and protection. A dynamic stable state of surface evolution was established, as the reviewer said.

The following is the replacement of the stability image in the manuscript, adding 3000 h stability data.

Fig. 2. | PEC properties of the β -Fe₂O₃ photoanode in simulated seawater.

a, Stability test of β -Fe₂O₃ (in 1 M KOH, 1 M KOH/0.5 M NaCl, 1 M KOH/saturated NaCl solution) and Sn/ β -Fe₂O₃ (in 1 M KOH/saturated NaCl solution) for 100 h. b, AC electrochemical impedance spectra of Sn/ β -Fe₂O₃ before and after the reaction. c, HAADF images of the Sn/ β -Fe₂O₃ photoanode after 100 h of reaction in 1 M KOH/0.5 M NaCl. d, Stability test of Sn/ β -Fe₂O₃ in 1 M KOH/0.5 M NaCl for 3000 h. e, Summary of the photoanode stability of PEC (simulated) seawater splitting over the years. Detailed information can be found in Supplementary Table 1.

The following is the description added to the text.

Lines 201–217 (Original manuscript lines 192):

Such characteristics make the stability rise continuously at first, then decline slowly, and finally maintain a stable range of fluctuations. At the initial stage of the reaction, the β -Fe₂O₃ on the surface is converted into FeOOH in situ (Fig. 3). FeOOH itself is an efficient electrocatalyst, so the photocurrent increased. Because of the infiltration corrosion of Cl⁻ along with the excessive surface reconstruction and the loss of Sn, the photocurrent density had a downwards trend after 700 h. It can be analyzed from Supplementary Fig. 10 that due to the electrocatalytic effect of surface FeOOH after 1000-h reaction, the on-set potential moved to the negative direction by approximately 0.1 V_{RHE}. However, the photocurrent density at 1.6 V_{RHE} was reduced with the influence of surface reconstruction and corrosion. Owing to the relatively uniform distribution of Sn in the whole film, reconstruction and corrosion were controlled within a certain range of β -Fe₂O₃ surface, instead of continuing to the deep layer. In addition, while the surface reconstruction was accompanied by the loss of Sn, there would also be a sedimentation equilibrium on the surface, and deposition would occur, which forms a dynamic balance of corrosion, metal loss, deposition and protection. A dynamic stable state of surface evolution was established after a period of time, so the photocurrent density tended to be stable after 1000 h.

2) Although the size (1 × 1 cm) of the photoanode is reported in the Methods, it would be more straightforward to report photocurrent density instead of photocurrent in Figure 2a and 2d.

Response: We appreciate the reviewer's comments. We unify the ordinates in the figure as the current density.

We have made corresponding modifications to Figures 2a and 2d.

Fig. 2. | PEC properties of the β -Fe₂O₃ photoanode in simulated seawater.

a, Stability test of β -Fe₂O₃ (in 1 M KOH, 1 M KOH/0.5 M NaCl, 1 M KOH/saturated NaCl solution) and Sn/ β -Fe₂O₃ (in 1 M KOH/saturated NaCl solution) for 100 h. b, AC electrochemical impedance spectra of Sn/ β -Fe₂O₃ before and after the reaction. c, HAADF images of the Sn/ β -Fe₂O₃ photoanode after 100 h of reaction in 1 M KOH/0.5 M NaCl. d, Stability test of Sn/ β -Fe₂O₃ in 1 M KOH/0.5 M NaCl for 3000 h. e, Summary of the photoanode stability of PEC (simulated) seawater splitting over the years. Detailed information can be found in Supplementary Table 1.

3) If possible, the current-potential curves of the photoanodes after the stability tests are suggested to be provided, as they provide meaningful information on the change of the onset potentials after the stability test.

Response: We are grateful for the reviewer's suggestions. We have added the current-bias curve of Sn/ β -Fe₂O₃ after 1000-h reaction. Due to the partial reconstruction of the surface, FeOOH was generated in situ. FeOOH could act as an electrocatalyst, and the on-set potential of the photoanode after the stability test moved towards the direction of low bias voltage. Due to the electrocatalytic effect of surface FeOOH after 1000-h reaction, the on-set potential moved to the negative direction by approximately 0.1 V_{RHE}. But the photocurrent density at 1.6 V_{RHE} was reduced with the influence of surface reconstruction and corrosion.

We added the photocurrent curve before and after the reaction as Supplementary Fig. 10:

Supplementary Fig. 10. j-V curves of Sn/ β -Fe₂O₃ before and after 1000 h of reaction.

The following is the description added to the text.

Lines 202–212 (Original manuscript lines 192):

At the initial stage of the reaction, the β -Fe₂O₃ on the surface is converted into FeOOH in situ (Fig. 3). FeOOH itself is an efficient electrocatalyst, so the photocurrent increased. Because of the infiltration corrosion of Cl⁻ along with the excessive surface reconstruction and the loss of Sn, the photocurrent density had a downwards trend after 700 h. It can be analyzed from Supplementary Fig. 10 that due to the electrocatalytic effect of surface FeOOH after 1000-h reaction, the on-set potential moved to the negative direction by approximately 0.1 V_{RHE}. However, the photocurrent density at 1.6 V_{RHE} was reduced with the influence of surface reconstruction and corrosion. Owing to the relatively uniform distribution of Sn in the whole film, reconstruction and corrosion were controlled within a certain range of β -Fe₂O₃ surface, instead of continuing to the deep layer.

4) Line 43, Refs 6 and 7 are not about β -Fe₂O₃, please double-check the accuracy of the reference list.

Response: Thanks for the reviewer's comment. Here is a citation error. We have added the correct and corresponding references here, and other corresponding reference numbers have also been changed.

References

Lines 325–329 (Original manuscript lines 298):

[20] Zhang, N. S. et al. Paving the road toward the use of β -Fe₂O₃ in solar water splitting: Raman identification, phase transformation and strategies for phase stabilization. *Natl. Sci. Rev.* **7**, 1059–1067, (2020).

[21] Li, Y. et al. Metastable-phase β -Fe₂O₃ photoanodes for solar water splitting with durability exceeding 100 h. *Chinese J. Catal.* **42**, 1992–1998, (2021).

REVIEWERS' COMMENTS

Reviewer #2 (Remarks to the Author):

The author's response has satisfied me, and their extension of the stability experiment from 1000 hours to 3000 hours further demonstrates the stability of the material in PEC. Therefore, I would like to recommend publishing this article in its current version in Nature Communications.

Reviewer #3 (Remarks to the Author):

The authors have adequately addressed my concerns and the manuscript can be published after fixing the following minor issues. No further review is required.

- 1) Ref 20 in the main text differs from the one in the response letter.
- 2) The stability data (1440 h) in Supplementary Table 1 needs to be updated since they provided longer stability data (3000 h).

Point-by-point responses for Nature Communications manuscript

(ID: NCOMMS-22- 49861A)

Manuscript Type: Article

Title: Long-term durability of metastable β -Fe₂O₃ photoanodes in highly corrosive seawater.

Author(s): Changhao Liu, Ningsi Zhang, Yang Li, Rongli Fan, Wenjing Wang, Jianyong Feng, Chen Liu, Jiaou Wang, Weichang Hao, Zhaosheng Li, Zhigang Zou

General response:

We sincerely thank the editor, editorial staff and all reviewers for their comments. The manuscript has been modified point-by-point after addressing all the suggestions as listed below.

(Our response is given in blue and the corrections in the revised manuscript are shown in red)

Point-by-point responses to Reviewer(s)

Reviewer #2:

The author's response has satisfied me, and their extension of the stability experiment from 1000 hours to 3000 hours further demonstrates the stability of the material in PEC. Therefore, I would like to recommend publishing this article in its current version in *Nature Communications*.

Response:

We thank the reviewer for the very positive assessment of our work, which have greatly helped us improve the quality of the manuscript.

Reviewer #3:

The authors have adequately addressed my concerns and the manuscript can be published after fixing the following minor issues. No further review is required.

Response: We are very grateful to the reviewer for the suggestions and comments, which leads to further improvements in the quality of our work. According to the comments, we have modified our manuscript discussion and corresponding responses.

1) Ref 20 in the main text differs from the one in the response letter.

Response: Thanks for the reviewer's comment. We apologize for the errors in references 5, 20, and 21 in the main text, and have made corrections accordingly.

References

Lines 350–351, and 391–395

[5] Nishiyama, H. et al. Photocatalytic solar hydrogen production from water on a 100-m² scale. *Nature* **598**, 304–307 (2021).

[20] Zhang, N. S. et al. Paving the road toward the use of β -Fe₂O₃ in solar water splitting: Raman identification, phase transformation and strategies for phase stabilization. *Natl. Sci. Rev.* **7**, 1059–1067 (2020).

[21] Li, Y. et al. Metastable-phase β -Fe₂O₃ photoanodes for solar water splitting with durability exceeding 100 h. *Chinese J. Catal.* **42**, 1992–1998 (2021).

2) The stability data (1440 h) in Supplementary Table 1 needs to be updated since they provided longer stability data (3000 h).

Response: Thanks for the reviewer's comment.

The data in Supplementary Table 1 has been updated (Supplementary Materials lines 118).

Supplementary Table 1. Comparison of photoelectrochemical OER (simulated) seawater splitting with different photoanodes.

Photoanodes	Light intensity	Bias potential	Current density	Stability	Ref.
Sn/ β -Fe ₂ O ₃	AM 1.5G	1.6 V _{RHE}	2.21 mA cm ⁻²	3000 h	this work
RhO ₂ /Mo-BiVO ₄	Full-arc xenon lamp ($\lambda > 300$ nm) with higher light intensity	1.0 V _{Ag/AgCl}	18 mA cm ⁻²	270 min	14
TiO ₂ @g-C ₃ N ₄ @CoPi	AM 1.5G	1.23 V _{RHE}	1.64 mA cm ⁻²	10 h	38

WO ₃ /g-C ₃ N ₄	AM 1.5G	1.23 V _{RHE}	0.73 mA cm ⁻²	1 h	39
Fe ₂ O ₃ /WO ₃	AM 1.5G	1.23 V _{RHE}	1 mA cm ⁻²	5 h	8
In ₂ S ₃ /ANP/RND	AM 1.5G	1.23 V _{RHE}	1.53 mA cm ⁻²	2 h	9
In ₂ S ₃ /In ₂ O ₃	AM 1.5G	0.981 V _{RHE}	~0.2 mA cm ⁻²	1000 s	40
Mg doped ZnO	AM 1.5G	0.5 V _{Ag/AgCl}	~1 μA cm ⁻²	5 h	41
MoB/BiVO ₄	AM 1.5G	1.23 V _{RHE}	4.30 mA cm ⁻²	70 h	42
NiMoO _x /BiVO ₄	AM 1.5G	1.23 V _{RHE}	3.0 mA cm ⁻²	190 h	43
Bi ₂ S ₃ /NiS/NiFeO/TiO ₂	300 W Xe lamp	1.23 V cell voltage	10 mA cm ⁻²	4 h	44
Bi _{0.6} Fe _{0.4} VO ₄ @CNTs	AM 1.5G	1.5 V _{Ag/AgCl}	~0.1 mA cm ⁻²	1 h	45